# Rheology of rounded mammalian cells over continuous high-frequencies

Gotthold Fläschner [1], Cosmin I. Roman[2], Nico Strohmeyer [1], David Martinez-Martin [1,3] & Daniel J. Müller [1✉]

Understanding the viscoelastic properties of living cells and their relation to cell state and morphology remains challenging. Low-frequency mechanical perturbations have contributed considerably to the understanding, yet higher frequencies promise to elucidate the link between cellular and molecular properties, such as polymer relaxation and monomer reaction kinetics. Here, we introduce an assay, that uses an actuated microcantilever to confine a single, rounded cell on a second microcantilever, which measures the cell mechanical response across a continuous frequency range ≈ 1–40 kHz. Cell mass measurements and optical microscopy are co-implemented. The fast, high-frequency measurements are applied to rheologically monitor cellular stiffening. We find that the rheology of rounded HeLa cells obeys a cytoskeleton-dependent power-law, similar to spread cells. Cell size and viscoelasticity are uncorrelated, which contrasts an assumption based on the Laplace law. Together with the presented theory of mechanical de-embedding, our assay is generally applicable to other rheological experiments.

[1] Eidgenössische Technische Hochschule (ETH) Zürich, Department of Biosystems Science and Engineering, Basel, Switzerland. [2] Eidgenössische Technische Hochschule (ETH) Zürich, Department of Mechanical and Process Engineering, Zürich, Switzerland. [3] Present address: The University of Sydney, School of Biomedical Engineering, NSW Sydney, Australia. ✉email: daniel.mueller@bsse.ethz.ch

Material properties of living cells are notoriously difficult to measure due to their dynamic and heterogeneous nature[1,2]. One key determinant of cell mechanical properties is the dynamic cortical cytoskeleton, which mainly consists of a tensioned actomyosin network, that is assembled and maintained by more than hundred different proteins[3,4]. To understand the contribution of the cortex to cell mechanics, one thus attempts to simplify the cellular complexity to physical quantities such as cortical thickness, tension, or length of filaments to cell viscosity and elasticity[5,6]. Hence, to lay the foundation for model building and improvement, the cellular viscoelasticity needs to be characterized. However, viscoelastic properties are frequency dependent as stress relaxation occurs at different timescales[7]. Passive and active rheology approaches characterize the viscoelasticity by the diffusion of particles on/inside a cell[8] and by measuring the cellular response to an active deformation[9], respectively. So far, low and intermediate frequency ($10^{-3}$–$10^2$ Hz) recordings predominate active rheology techniques because these frequencies are easily approachable. Only recently active rheology techniques have started to cover high-frequency regimes (»10 kHz)[10,11], which all have to overcome the technological problem of introducing mechanical resonances that can obscure the mechanical response of the biological system. In principle, atomic force microscopy (AFM)[10], which is possibly the most widely applied tool to characterize the morphological and mechanical properties of biological systems, could characterize rheological properties at higher frequencies because the probing AFM microcantilevers can show relatively high resonance frequencies. However, to characterize soft biological systems such as living cells requires using soft microcantilevers, which relatively low resonance frequencies in liquid ($f_{cant} \approx 10$ kHz) limits their application for high-frequency measurements. An exception are soft, high-speed cantilevers with $f_{cant} \approx 100$–$1000$ kHz[12], which small sizes are rather limited to locally probe the heterogeneous morphological and mechanical properties of living cells. We here overcome both limitations by introducing a nanotechnological assay that sandwiches a rounded mammalian cell between two parallel microcantilevers, one of which being photothermally actuated. Using this so-called parallel-plate assay[13], we analyze how mechanical perturbations in a continuous frequency range of ≈ 1–40 kHz propagate through the entire cell cortex. The analysis enabled by a theoretical framework decouples the mechanical properties of cantilevers and cell to extract the high-frequency response of the entire cell. This rheological response of the cell is recorded within seconds and provides a snapshot of the cell state to which cellular components, such as actin, myosin or intermediate filaments[14] contribute. We find the rheological response of the rounded cell to follow a double power-law behavior. By relating the measurements with the cell mass, we find that the cell elasticity and viscosity are independent of cell size and volume. The finding stays in apparent contrast to an interpretation of the Laplace law that cell pressure and/or cortex tension depend on cell size and volume.

## Results

**Monitoring the stiffening of a rounded cell.** Our nanotechnological assay characterizes the mechanical properties of a single, rounded cell that is being sandwiched between two microcantilevers at cell culture conditions (Fig. 1a, b)[15,16]. Using a blue laser the master microcantilever is actuated by photothermal force over a wide range of frequencies to exert mechanical force on the sandwiched living cell. The cell transduces the force to the second slave microcantilever, which movements are detected by a red laser. Thus, characterizing the force transduction between both microcantilevers allows to extract the mechanical properties of the cell. The assay, which averages sub-cellular heterogeneities that usually spread the measured mechanical response[17], analyzes the mechanical behavior of the entire cell and allows to compare cellular populations. To start a measurement, the microcantilever mounted on a holder, which can be freely moved in three dimensions, is used to pick up and to measure the total mass $m_c$ of a rounded HeLa cell[15]. The cell is then sandwiched between master and slave microcantilever and characterized rheologically in a parallel-plate assay (Fig. 1a). To describe the coupled system of microcantilever–cell–microcantilever, we used a lumped-element model that simplifies the spatially distributed experimental setup into a configuration of two microcantilever masses $m_m$ and $m_s$, to which also the cell mass $m_c$ contributes (Supplementary Note 1). Both microcantilever masses are coupled to the cantilever chip via the transfer functions $g_m(f)$ and $g_s(f)$ and to each other via the mechanical transfer function $g_c(f)$ of the cell (Fig. 1c).

First, we tested our experimental setup qualitatively. A rounded living HeLa cell was sandwiched between both microcantilevers and the master microcantilever was oscillated at an amplitude of ≈10 nm through frequencies sweeping from ≈1 to 38 kHz, while measuring the amplitude and phase response of the slave microcantilever (Fig. 1d). The measurement did not compromise cell viability (Supplementary Fig. 1). In the absence of the cell the response of the slave microcantilever dropped to ≈4% (Supplementary Fig. 2), which shows that the sandwiched cell dominates the mechanical coupling between both microcantilevers and that the penetration depth $\delta_p$ of the mechanical waves inside the cell is magnitudes larger than that in water. Upon sandwiching a cell this coupling of the medium must drop below 4%, as it is largely replaced by the cell. Recording a single frequency sweep of the sandwiched cell at 200 Hz resolution took only ≈4 s as little data averaging is needed due to the high signal-to-noise ratio of the setup (Supplementary Fig. 3). Next, we wanted to rheologically monitor the mechanically stiffening of the cell. We thus added 1% of the chemical crosslinker glutaraldehyde to the medium and acquired amplitude and phase response curves over the time range of 25 min (Fig. 1d). Crosslinking increased the mechanical transduction of the cell, which resulted in a higher amplitude and a smaller phase delay of the slave microcantilever. Upon modeling the coupled microcantilever–cell–microcantilever system by an effective spring-dashpot-system actuated by a photothermal force[18] (Supplementary Fig. 4), the crosslinking process stiffening the cell and thus changing the cell mechanical response could be mimicked by gradually increasing the spring constant of the cell (Fig. 1e). While the experiments show that the assay measures the overall stiffness of the cell, the cell mechanical properties can only naïvely be described by a single spring.

**Mechanical de-embedding: resonant and sub-resonant rheology.** To extract the cell mechanical properties (Fig. 1c), we solved the equations of motion of the coupled microcantilever–cell–microcantilever system (Supplementary Note 1). The cellular transfer function $g_c(f)$ mechanically coupling both microcantilevers can be written as:

$$g_c(f) = \frac{\chi_{MS}(f)}{\chi_{MM}(f) \cdot \chi_{SS}(f) - F_m/F_s \chi_{MS}^2(f)} F_m \qquad (1)$$

where $\chi_j(f) = A_j(f)e^{-i\phi_j(f)}$ represents the transfer function of an oscillator with $A_j(f)$ being the amplitude and $\phi_j(f)$ the phase response measured in three configurations $j \in [MM, MS, SS]$ (Fig. 2a). $F_m$ and $F_s$ are the effective forces actuating the master and slave microcantilever, respectively (Fig. 1c). This mechanical coupling is independent of the cantilever properties such as for instance their inertia (Supplementary Note 1), while the inertia of the

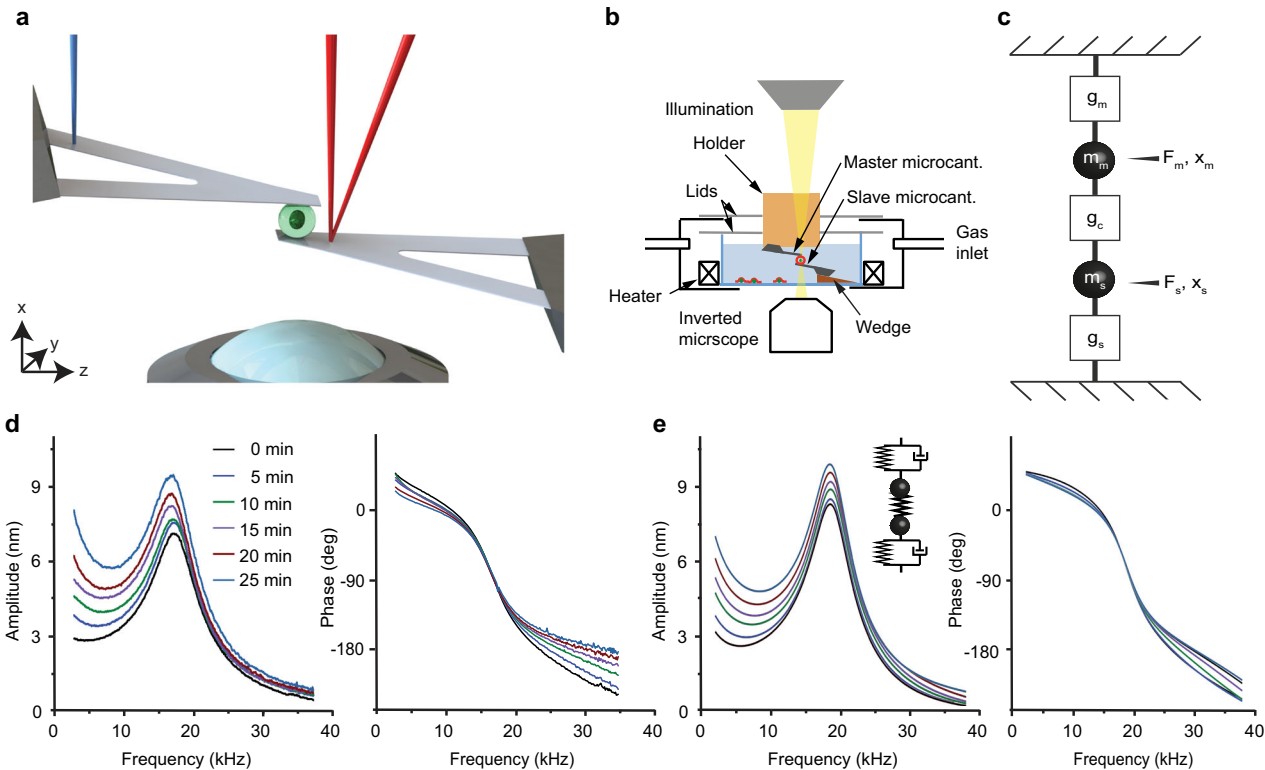

**Fig. 1 Working principle and conceptual proof of high-frequency rheology over a continuous frequency range. a** A rounded cell (green) confined between two parallel microcantilevers. A blue laser actuates the master microcantilever while a red laser reads the motion of the slave microcantilever and measures the cell mechanical properties. **b** For morphological characterization the setup is combined with an optical microscope. A chamber controlling humidity, temperature, and gas atmosphere maintains cell culture conditions during the experiments. To arrange both microcantilevers in a parallel-plate assay, the slave microcantilever is mounted on a wedge. **c** Lumped-mass model of configuration shown in **a**. The motions of both microcantilevers, which have effective masses $m_m$ and $m_s$, are determined by transfer functions $g_m(f)$ and $g_s(f)$, which describe their respective coupling to their cantilever chips. Both cantilevers are coupled to each other via the transfer function of the cell $g_c(f)$. **d** Amplitude (left) and phase (right) measurements of a rounded HeLa cell exposed to glutaraldehyde and hardening over 25 min. At 0 min, glutaraldehyde was added to the medium to a final concentration of 1% (vol/vol). **e** Simulation of amplitude (left) and phase (right) measurements in a lumped-mass system (inset). Microcantilevers are modeled as Kelvin–Voigt elements, the cell as a simple spring (Methods). The curves are generated for a gradual increase of the cellular spring constant.

cytosolic fluid becoming important at frequencies >400 kHz at our cantilever oscillation amplitudes[19]. The application of Eq. 1 needs to consider that the microcantilever movements are not actuated and readout at the same positions as assumed in the lumped-mass model (Fig. 2b). Shifting the position of the actuating blue laser $z_{blue}$ from the free end towards the base of the microcantilever alters the response function (Fig. 2c, Supplementary Fig. 5) as described by finite elements (FEM) simulations (Supplementary Fig. 6). To test the extraction method summarized in Fig. 2d, we simulated the whole microcantilever–cell–microcantilever setup using FEM (Fig. 2e, Methods). We assumed five different functions for the dynamic modulus of the cell cortex and generated the amplitude and phase response curves of the coupled system (Supplementary Fig. 7a, b). From these curves, we extracted the cellular response function (Fig. 2f, Supplementary Fig. 7c) with only minor deviations (<2%) compared to the response function determined from simulating the force-deformation of the cell without cantilevers (Fig. 2g). As additional experimental control, we tested our extraction method using micrometer-sized alginate beads, from which we attained rheological properties agreeing with literature values (Supplementary Fig. 8). As the extraction method decouples the mechanical properties of the microcantilevers from that of the cell, we term this framework mechanical de-embedding.

**Cytoskeleton-dependent rheological properties of rounded cells.** Next, we characterized the mechanical properties of single

HeLa cells (Fig. 3). Rounded cells were picked up, weighted, placed into the parallel-plate configuration and compressed by ≈1 μm. After 5 min, to allow the cell to adapt to the confinement (Supplementary Fig. 9), we recorded frequency-sweeps using all three configurations (Fig. 2a), which took ≈15 s (Fig. 3a, Supplementary Fig. 10). A reduction of the actuation amplitude left the response of slave cantilever unaltered except for its amplitude, which was reduced over all frequencies by the same factor as the actuation (Supplementary Fig. 11). This linear response, which indicates that the cell responds linearly to the oscillatory compression of the cantilever, is prerequisite for the description of the cellular behavior with the transfer function $g_c(f)$. We then used our mechanical de-embedding framework to extract $g_c(f)$ from the amplitude and phase response curves. From $g_c(f)$ we then applied the solid-shell–liquid-core (SSLC) model to calculate the storage $E'_{cort}$ and loss $E''_{cort}$ moduli of the cell cortex (Methods, Supplementary Note 2), which describe the mechanical energy stored and lost per cortex volume, respectively (Fig. 3b). The SSLC model was found appropriate for round cells[20]; however, the choice of a different model can impact the moduli's magnitudes, as illustrated by the Hertz model, which results in significantly lower moduli[20] as a consequence of assuming considerably larger volumes in which energy can be stored/lost. To calculate $E'_{cort}$ and $E''_{cort}$ within the SSLC model, we assumed a cortical thickness of $h_{cort} \approx 200$ nm[21] and simulated the geometry

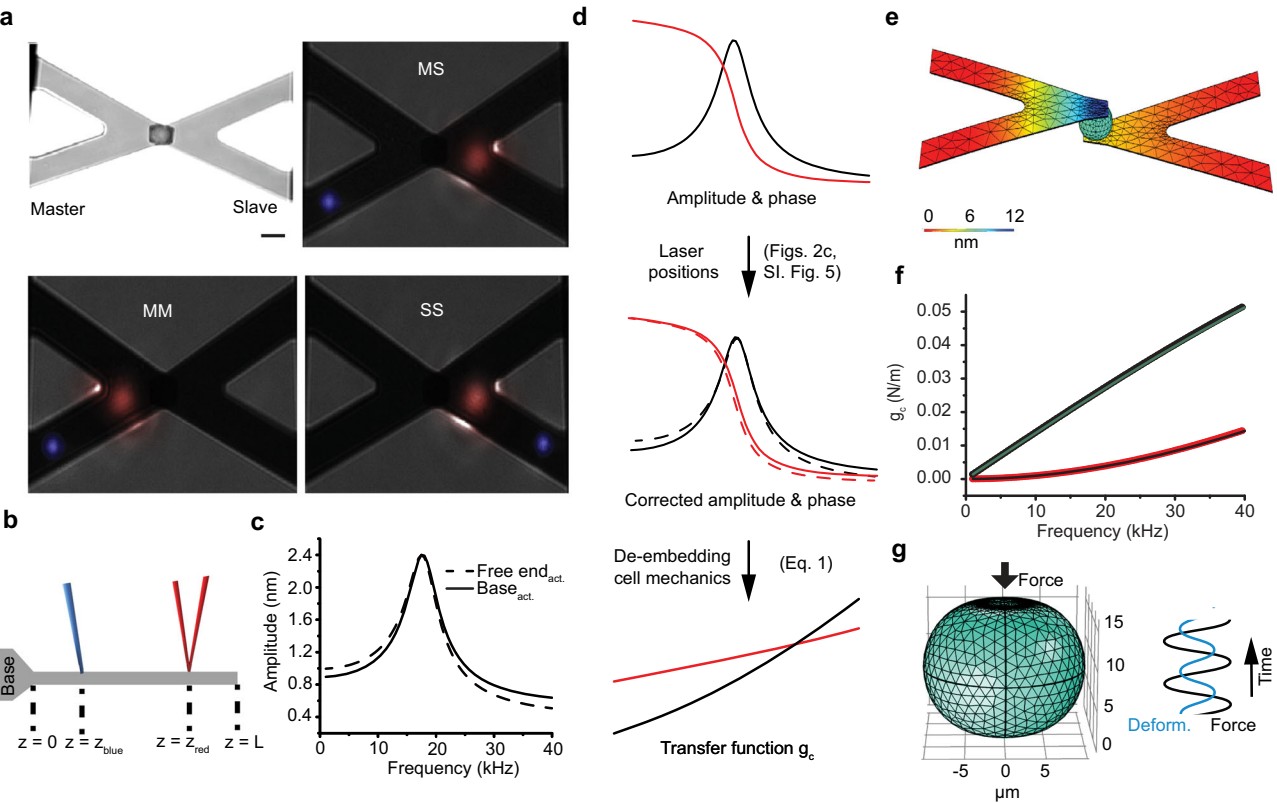

**Fig. 2 Extracting viscoelastic properties by mechanical de-embedding framework. a** To extract the mechanical properties of the cell, the setup is characterized using three sets of laser positions MM, MS, and SS. Top left, differential interference contrast (DIC) image of a rounded HeLa cell confined between master (left) and slave (right) microcantilever. Scale bar, 10 μm. DIC images (false colored) show the actuating blue laser having a spot size of 6 μm and red readout laser having a spot size of 21 μm. **b** Diagram showing the position of blue actuating and red readout laser spots on the cantilever. **c** Numerically acquired amplitude responses of a triangular cantilever actuated and readout at the free end ($z_{blue} = z_{red} = 0.9L$, dashed line) and actuated at the base end and readout at the free end ($z_{blue} = 0.1L$, $z_{red} = 0.9L$, black line). Together with the corresponding phase curves (Supplementary Fig. 5), a correction for the laser position was extracted (Methods, Supplementary Fig. 6). **d** Procedure to extract the mechanical properties of the cell. Top, experimental acquisition of amplitude (black) and phase (red) response curves of the microcantilever. Middle, mathematically shifting the position of the blue laser actuating the cantilever to coincide with the point of cantilever readout gives a corrected amplitude (black, dashed) and phase (red, dashed). Bottom, applying the transfer function $g_c(f)$ to extract the storage (red, $E'_{cort}$) and loss (black, $E''_{cort}$) moduli. **e** Model used for FEM simulations. Microcantilever dimension, stiffness, damping, and cell size are taken from the experiment. Colors encode deflections along the microcantilever actuated at 19 kHz (Methods). The model generates amplitude and phase responses to test the mechanical de-embedding framework (Supplementary Fig. 7a, b). **f** Extracting $g_c$ (black data points imaginary part, red data points real part) by applying the method summarized in **d** to amplitude and phase response curves (Supplementary Fig. 7a, b) generated in **e**. The results are compared to $g_c(f)$ calculated from simulating of a deformed cell excluding both cantilevers (thin green and black lines). Only deviations <1% are found for this particular rheological (Maxwell) model (other rheological models shown in Supplementary Fig. 7c). **g** Simulation of a single cell being deformed by a force. The relationship between the time-dependent force and deformation allows to determine $g_c(f)$.

of the sandwiched cell using the compression distance $\Delta$ and the cell radius $R$, which was derived with $R = \sqrt[3]{\frac{3}{4\pi} m_c/\rho_c}$ from the cell mass $m_c$ and cellular density $\rho_c$. This estimation of $R$ is very accurate ($\Delta R \approx 150$ nm) due to the precise cell mass measurement and small cellular density variations[22]. We confirmed our derived cell geometry by super-resolution optical microscopy (Supplementary Fig. 12). The extracted rheological data of rounded HeLa cells was best fitted applying a double power-law model (Supplementary Table 1) that defines the dynamic modulus

$$E^*_{cort}(f) = E'_{cort}(f) + iE''_{cort}(f) = A\left(if/f_0\right)^\alpha + B\left(if/f_0\right)^\beta \quad \text{(Eq. 2)}$$

and has been used to describe rheology[8,23] of spread cells before. $A$ and $B$ are scaling factors, $\alpha$ and $\beta$ low- and high-frequency exponents, $f_0$ a normalization constant set to 1 kHz, and $i$ the imaginary unit. A striking feature of the cellular viscoelastic response was a crossing frequency of storage and loss moduli at >40 kHz (Fig. 3b). Our observation differs from results of

magnetic twisting cytometry of human airway smooth muscle cells at ambient conditions[24]. However, a crossing frequency of both moduli similar to our results had been previously found by locally indenting 3T3 fibroblasts at ambient conditions using spherical high-speed AFM probes[10]. As our single-cell parallel-plate assay provides cell culture conditions, we conclude that this crossing reflects the mechanical properties of entire cortex at physiological conditions. One may further speculate whether cell type specific differences in the actomyosin structures or the experimental approach and condition may have caused smooth muscle cells to behave differently compared to fibroblasts and HeLa cells.

To further investigate the role of the actomyosin cortex, we perturbed the cortex using CK666 to inhibit Arp2/3, which leads to a more parallel arrangement of the actin filaments[25], or LatA to inhibit the polymerization of filamentous actin by sequestering monomeric actin[26] (Fig. 3c, d). The rheological response of the chemically perturbed HeLa cells again showed a double power-law

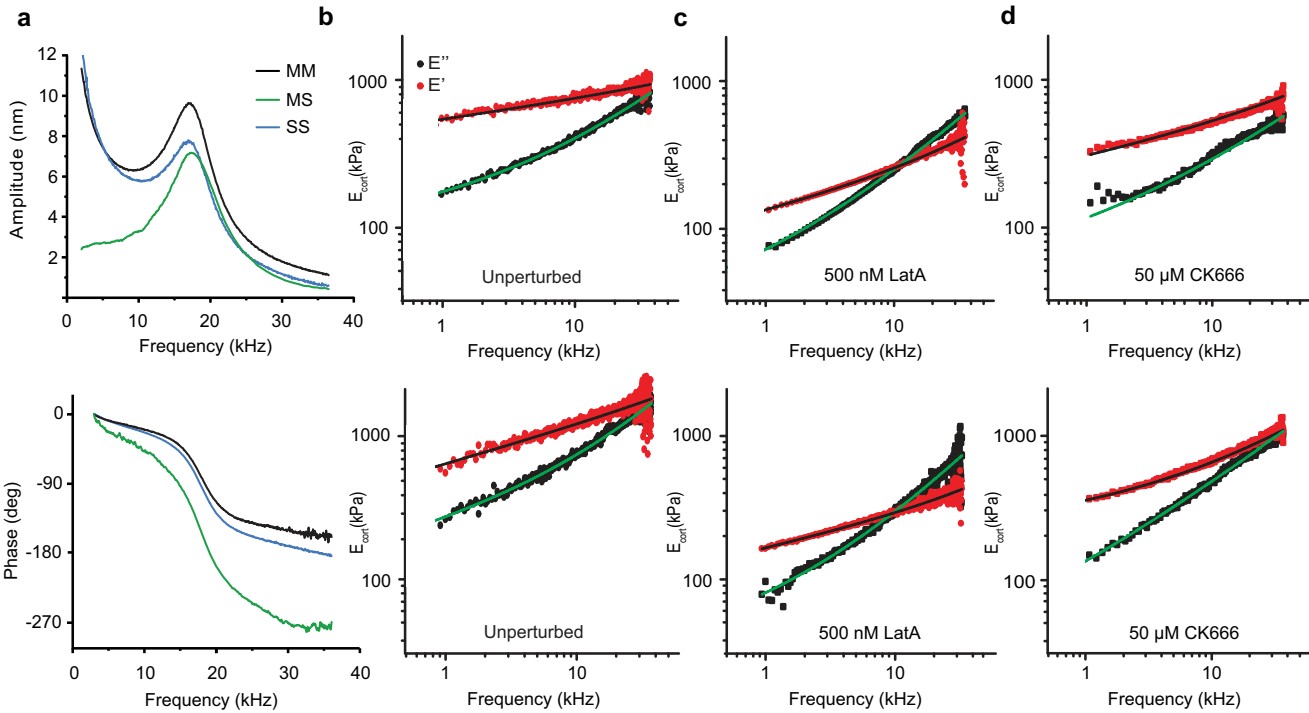

**Fig. 3 High-frequency rheology on unperturbed and F-actin perturbed HeLa cells. a** Response curves of an unperturbed HeLa cell measured in configuration MM, MS, and SS (Fig. 2a). MM shows the response curve of the actuated master and SS of the actuated slave microcantilever. MS shows the slave microcantilever responding to the force of the photothermally actuated microcantilever transmitted through the cell. The raw data of the corresponding functions $\chi_{MM}$, $\chi_{MS}$, and $\chi_{SS}$ is displayed in Supplementary Fig. 10. **b** Cell rheological responses measured for two single HeLa cells. The extracted storage $E'_{cort}$ (red) and loss $E''_{cort}$ (black) moduli of HeLa cells can be fitted using a double power-law behavior (black and green lines, respectively). At frequencies >35 kHz, the microcantilever response is noisier leading to the scattering of the data. **c** Cell rheological responses measured for two HeLa cells perturbed with 500 nM LatA. The frequency at which $E''_{cort}(f) = E'_{cort}(f)$ descreases considerably, which describes the cells to increase viscosity. **d** Cell rheological responses measured for two HeLa cells perturbed with 50 μM CK666. The frequency at which $E''_{cort}(f) = E'_{cort}(f)$ increases, indicating a more elastic response of the cell.

behavior. The mean storage $E'_{cort}$ and loss $E''_{cort}$ moduli of native unperturbed and chemically perturbed HeLa cells are displayed in Fig. 4a with the averaged parameters $A$, $B$, $\alpha$, and $\beta$ given in Supplementary Table 2. Perturbing the actin polymerization by LatA caused $E'_{cort}$ to cross $E''_{cort}$ at much lower frequencies of ≈10.5 kHz compared to unperturbed cells, which describes the HeLa cell to become more viscous. Whereas $\alpha$ and $\beta$ altered only slightly, the scaling factor $A$ dropped by a factor 2.6, which compares well with previous static, parallel-plate measurements of rounded cells[27]. However, compared to values previously gained upon locally indenting spread 3T3 fibroblasts[10], our values differ, which may be accounted to the fundamentally different structures and mechanical properties of the cortices of rounded and spread cells, the different cell type, or the different theoretical models applied to describe the AFM-based local indentation into a heterogeneous and thin cell cortex[2]. We conclude that the considerable drop in $A$ revealed in our rheology experiments describes the deconstruction of the actomyosin cortex of the entire cell, whereas the unaltered power-law exponents $\alpha$ and $\beta$ indicate that the fluctuations of single actin filaments governing the high-frequency response[28] remain largely unaffected.

Upon inhibiting Arp2/3 in HeLa cells, $E'_{cort}$ equaled $E''_{cort}$ at much higher frequencies, which describes the HeLa cell to increase elasticity. Considering that $A$ and the power-law exponents $\alpha$ and $\beta$ dropped considerably in the presence of CK666, the increased elasticity may appear counterintuitive. However, the scaling factor $B$ increased drastically at the same time. It has been described that extending the length of the actin filaments decreases $E'_{cort}$[6,29] and increasing the cortex tension

lowers the frequency dependency of the mechanical properties[30,31]. Hence, the results are in good agreement with theories describing the mechanical behavior of pre-tensed filaments[32] and experiments showing that CK666, which reduces the amount of short filaments in branched actomyosin networks[33], increases cortex tension[21]. Additionally, it may be assumed that the altered architecture of the cortex changes how it relaxes in response to mechanical stress[34], which would affect the exponential factors of the power-law behavior.

**Testing the generality of the Laplace law.** Next, we rheologically tested whether the cellular viscosity and elasticity depend on cell size, such as it can be interpreted from the law of Laplace[35]. The law, which is applied to describe the mechanical properties of rounded cells since decades[36], relates tension $T$, pressure $P$, and radius $R$ of a spherical cell by $T = P \cdot R/2$ (Fig. 4b). Thus, if rounded cells would universally adjust tension and pressure with their diameter, such regulation would be reflected in their mechanical properties (Supplementary Fig. 13, Supplementary Note 3). To investigate this regulation, we compared the mechanical properties of rounded HeLa cells naturally varying in size. We did not observe any correlation between the cell radius and the storage and loss moduli over all frequencies tested from ≈1 to 40 kHz (Fig. 4c, d) nor for the power-law parameters $A$, $B$, $\alpha$, and $\beta$ describing the cell rheological properties (Supplementary Fig. 14). We then employed stimulated emission depletion (STED) nanoscopy[37] to test whether the thickness of the actomyosin cortex correlates to the mechanical response of

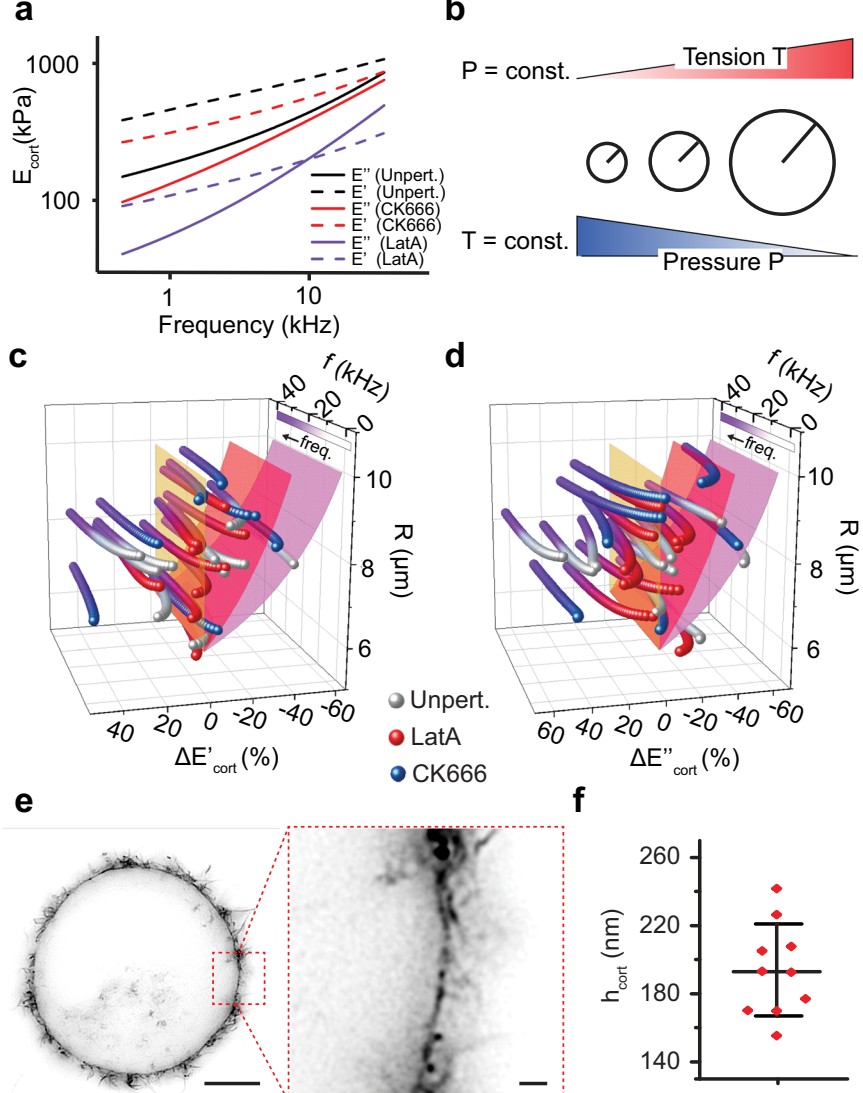

**Fig. 4 Rounded HeLa cells maintain no universal pressure or tension. a** Average power-law behavior of unperturbed HeLa cells (black) in the absence and in the presence of 500 nM LatA (purple) or 50 µM CK666 (red) ($n = 10$ biologically different cells for each condition), corresponding mean values and standard deviations are given in Supplementary Table 2. The storage modulus $E'_{cort}$ is plotted as a dotted line and the loss modulus $E''_{cort}$ as a through line for each condition. **b** Applying Laplace's law to a population of rounded cells gives two simple scenarios: i) bigger cells are stiffer, if a universal pressure exists, or ii) bigger cells are softer, if a universal tension exists. **c**, **d** Relative variations of storage and loss moduli of unperturbed (purple-gray), LatA perturbed (purple-red), and CK666 perturbed (purple-blue) HeLa cells plotted against the cell radius $R$ and frequency $f$. Negative and positive relative moduli describe the cell to become softer and stiffer, respectively. Frequency-dependent variations of both moduli are best visualized by the yellow reference plane. The hypothesis of no correlation with $R$ could not be rejected, as the $t$-test resulted in $p$-values of 0.95 and 0.92 (Methods). Red and pink planes indicate hypothetical dependencies proportional to $1/R$ (constant pressure) and $1/R^2$ (constant tension), respectively. An independency of cell size is found for every individual parameter of the power-law $A$, $B$, $\alpha$, and $\beta$ (Supplementary Fig. 14). **e** STED super resolution nanoscopy of paraformaldehyde-fixed HeLa cells with SiR-actin stained F-actin. Scale bars, 5 µm and 500 nm (inset). **f** For each of the 10 HeLa cells imaged by STED nanoscopy, we fitted 10 line profiles of the SiR-actin signal, to extract the cortical thickness $h_{cort}$ (194 ± 27 nm, mean ± S.D.). The center bar indicates the mean, the whiskers show the standard deviation. The diameter and cortex thickness measured for each cell by STED microscopy is shown in Supplementary Fig. 15.

the cell (Fig. 4e, f, Supplementary Fig. 15). The cortex thickness of ≈194 ± 27 nm, which varied up to 36%, did not correlate with cell size. Since the transfer function $g_c(f)$ was converted into storage and loss modulus under the assumption of a 200 nm thick cortex, this thickness variance could account for ≈40% variance of the moduli (Supplementary Fig. 16). Yet, within the storage and loss moduli we find a variation of up to ≈90% and ≈100%, respectively (Fig. 4c, d). We thus conclude that the variation of the viscoelastic parameters does not correlate with the cell radius, which indicates that rounded cells do not regulate pressure or tension in feedback to cell size and volume.

## Discussion

In summary, we have developed a technological assay to measure the spatially averaged, viscoelastic properties of single, rounded mammalian cells over continuous high-frequency ranges. The rheological assay, which includes mass measurements and optical microscopy, probes the properties of an entire single cell and thus prevents the measurements from becoming scattered by subcellular heterogeneities such as they occur in conventional AFM- or probe-based indentation experiments[2]. Upon applying the theoretical framework of mechanical de-embedding we overcome the restrictions that prevent measuring the rheological properties

regardless of the resonance frequency of an indenting probe. Principally, the theoretical framework is applicable to extend the frequency range of other rheology methods[38], which however, must be modified to measure continuous frequency ranges and to reveal the full rheological response of a sample. Upon characterizing the mechanical behavior of unperturbed and actin perturbed HeLa cells our assay confirms that rounded cells obey a power-law behavior[39], which has been intensively debated[40,41]. At the small mechanical amplitudes applied, the cell mechanical response strongly depends on the actomyosin cortex. Disrupting the actin polymerization lowers the crossing frequency up to which the cell stores more of the supplied mechanical energy than it dissipates to 10.5 kHz. Perturbation of the filamentous branching within the cortex, increases this crossing frequency, although the cortex could generally store less energy. This observation highlights that characterizing the parameters $A$, $B$, $\alpha$, and $\beta$ of the cell mechanical moduli in response to specific chemical or genetic perturbations can provide unique possibilities to learn about the contribution of cell state, cytoskeleton or compartments. The characterization of short-time scale processes, such as polymer relaxation and monomer-reaction kinetics, is important to understand how the long-time scale processes emerge from these, improve theoretical frameworks[10,14] and develop mechanistic, bottom-up models that account for the complexity of the cells interior. Furthermore, upon rheologically testing a possible generality of the law of Laplace, we found no correlation of cell size or volume with cell pressure or tension. This finding implies that cells within a population do not regulate pressure or tension in respect to cell size. In principle, our developed method can be used to explore higher force regimes and combined with other assays, such as passive, bead-based microrheology. Additionally, optical super-resolution microscopy may be implemented into our rheological assay to correlate cell mechanical response, morphology, and state. This would help to attain a better understanding of how the filamentous architecture, compartments, and components of living cells contribute to mechanobiological processes[38,39] and of the non-equilibrium contributions of active matter in general[42,43].

## Methods

**Cell culture**. Wild-type HeLa Kyoto cells (kindly provided by A. Hyman, Dresden) were cultured in phenol red-free, high-glucose Dulbecco's modified Eagle's medium (DMEM), supplemented with 1 mM sodium pyruvate, 4 mM GlutaMAX (Gibco Life Tech.), and 10% (v/v) fetal calf serum (Sigma Aldrich). The medium contained 100 units ml$^{-1}$ penicillin and 10 µg ml$^{-1}$ streptomycin (Gibco Life Technologies). Cells were cultured in T-25 flasks at 37 °C with 5% $CO_2$ in the atmosphere. Upon reaching a confluency of 70%, cells were split. Cells were used for a maximum of 25 passages. The cell line was regularly tested for mycoplasma contamination using a PCR kit (ATCC® 30-1012 K⁻).

**Cantilever calibration, cleanliness, and functionalization**. Cantilevers (NP-O Type A, Bruker) having a nominal Q-factor of 3 and spring constant of ≈0.35 N m$^{-1}$ were screened for the best photothermal response and calibrated using the Sader method[44,45]. For functionalization, cantilevers were cleaned in 95% sulfuric acid for 3 min at room temperature. During this time, the cantilevers were gently moved in the acid. Then, the cantilevers were rinsed three times with ultrapure water (≈18 MΩ cm$^{-1}$) and dried by placing their chips onto precision wipes (Kimtech Science). After this they were placed for 10 min in a UVO-cleaner® (Jelight Company Inc.). After UVO treatment, the cantilevers were incubated in 50 mg ml$^{-1}$ Concanavalin A (ConA, Calbiochem) for 1 h at 37 °C. Subsequently the cantilevers were rinsed with phosphate-buffered saline solution and immediately used to attach cells.

**System setup**. The device combining the photothermal actuation of micro-cantilevers and optical microscopy was used as described[15]. The device was supplemented with a z-piezo stage (CellHesion® module, JPK Instruments), which allowed precisely confining the cell between both cantilevers. The lower, slave cantilever bearing chip was fixed with vacuum grease onto a CNC milled peek wedge with a 10° angle and dimensions of 3.30 × 3.00 × 0.75 mm³ (W × L × H). The angled wedge allowed the readout laser to be reflected from the slave cantilever into the photodiode. The wedge was fixed with vacuum grease to a Petri dish. The device was combined with an inverted optical microscope (Zeiss, Axio Observer)

equipped with a OrcaFlash 4.0 camera (Hamamatsu) and a 20x plan apochromat objective with a 0.8 NA (Zeiss). All experiments were conducted in a controlled environmental system[16] to maintain cell culture conditions, temperature control (37.0 °C), and pH adjustment using a humidified gas mixture based on synthetic air containing 5% $CO_2$.

**Cell mass measurements**. Cell mass measurements were conducted as previously described[15]. Briefly, cells were grown in 6-well plates, then trypsinized. The cells were allowed to recover in cell culture medium (see Methods) for 30 min from trypsin treatment, before they were seeded into a Petri dish (Ibidi-IbiTreat). To attach a rounded HeLa cell to a cantilever, a ConA-functionalized cantilever was lowered onto a rounded cell until deforming the cell by ≈0.35 µm, maintained at constant height for 2 s. Afterwards, cantilever and attached cell were raised 200 µm above the dish bottom. The vertical movement completely detached the cell from the Petri dish and the distance was sufficient to avoid hydrodynamic coupling of the cantilever with the bottom of the dish. A sweep of the laser modulation frequency was performed between 1 kHz and 40 kHz, to determine the natural frequency of the cantilever before and after cell attachment. The cantilever amplitude was recorded to calculate the effective actuation force of the photothermal actuation (Supplementary Fig. 4).

**Cell mechanics measurements**. Cellular mechanics were characterized following cell mass measurements. Before bringing a cell, which adhered to the master cantilever, in contact with the slave cantilever, the very free ends of both cantilevers were brought into contact. By recording a force–distance curve for the master cantilever, the distance of both cantilevers was calibrated. In the following this distance was necessary to measure the height and compression of the sandwiched cell. From the point of contact with the slave cantilever, the master cantilever was raised by 25 µm. Then it was positioned in $x$–$y$, such that the attached cell was above the free end of the slave cantilever. The master cantilever was approached to the slave cantilever until the rounded cell was compressed by ≈1 µm. After 5 min, which allowed the cell to adapt to the confinement (Supplementary Fig. 9), we recorded frequency-sweeps in all three configurations (Fig. 2a). In every laser configuration, a microscopy image was taken. After the measurement, the master cantilever was moved away from the slave cantilever. A frequency-sweep was performed on the slave cantilever with the initial laser positions.

**Transfer function analysis**. First, the deflection sensitivity that was recorded prior to a mass measurement was used to convert the recorded voltages to amplitudes. The deflection sensitivity of both cantilevers was found to be identical in this setup (Supplementary Fig. 17). Then, the phase response curves recorded in three different laser configurations were corrected for a linear background, originating from the time delay introduced by the processing electronics. For this, the experimental phase response curve (Fig. 3a) of the single cantilever was fitted with the equation of a driven and damped harmonic oscillator having an additive linear phase background[46]: $\phi(f) = \text{atan}\left(\frac{f f_{cant}}{Q(f_{cant}^2 - f^2)}\right) + b_1 f + b_2$. Here, $f_{cant}$ is the natural resonance frequency and $Q$ the Q-factor describing the damping of the cantilever. The linear contribution $b_1 f$ and the offset $b_2$ were subtracted from the data. The amplitude response curves were fitted with $A(f) = \frac{1}{4\pi^2} \frac{F_0/m^*}{\sqrt{\left(f_{cant}^2 - f^2\right)^2 + \left(\frac{f_{cant} f}{Q}\right)^2}}$ in the center region of the natural resonance frequency, where $F_0$ is a constant force and $m^*$ the effective cantilever mass. From this, the effective, frequency-dependent actuation force $F(f)$ was extracted[17] (Supplementary Fig. 4). Next, the dimensions of the cantilever were used (Supplementary Fig. 6) to correct the amplitude and phase response curves. These corrections were derived from the FEM simulation for each laser position and cantilever length, based on microscopy images. Together with the effective actuation forces, the corrected amplitude and response curves were inserted into Eq. 1, to extract the transfer function $g_c(f)$. To account for the geometry of the cell and extract the storage and loss moduli from $g_c(f)$, we used a Comsol model (Supplementary Note 2 and Methods), which uses the cell radius $R$ and the thickness $h_{cort}$ of the cell cortex as parameters. For $h_{cort}$ we used 200 nm (Fig. 4f), whereas $R$ was calculated from the cell mass $m_c$ according to $R = \sqrt[3]{\frac{3}{4\pi} m_c/\rho_c}$, with $\varrho_c$ representing the density of the cell taken from ref. [15].

**Data fitting and statistical analysis**. Separating the double power-law $E_{cort}^*(f) = A(if/f_0)^\alpha + B(if/f_0)^\beta$ in real ($E'_{cort}$) and imaginary parts ($E''_{cort}$) leads to expressions:

$$E'_{cort} = A \cdot \cos\left(\frac{\pi\alpha}{2}\right) \cdot \left(\frac{f}{f_0}\right)^\alpha + B \cdot \cos\left(\frac{\pi\beta}{2}\right) \cdot \left(\frac{f}{f_0}\right)^\beta \qquad (2)$$

$$E''_{cort} = A \cdot \sin\left(\frac{\pi\alpha}{2}\right) \cdot \left(\frac{f}{f_0}\right)^\alpha + B \cdot \sin\left(\frac{\pi\beta}{2}\right) \cdot \left(\frac{f}{f_0}\right)^\beta \qquad (3)$$

Applying an unrestricted global fit (Origin 2018b, OriginLab Cooperation) to Eq. 2 and Eq. 3 were simultaneously fitted to the two sets of data, which corresponded to the imaginary and real parts of $g_c(f)$. The fits were performed for

every single cell using a two-dimensional optimization that iterated using the Marquart–Levenberg–Algorithm. The obtained values result from averaging the single cell parameters (Supplementary Table 2). To test the correlation of the fitting parameters $A$, $B$, $\alpha$, $\beta$ with the cell radius a Pearson coefficient was calculated.

To confirm the adequacy of the double-power law for the extracted data, the distance measures $\ell^{1*}$ and $\ell^{2*}$ were used, that asses how well the fit captures the data. The distance measures are defined as follows:

$$\ell^{1*} = \sum_{j=0}^{N} \mathrm{Abs}\Big(\mathrm{Re}(y_j - \hat{y}_j)\Big) + \mathrm{Abs}\Big(\mathrm{Im}(y_j - \hat{y}_j)\Big) \quad (4)$$

$$\ell^{2*} = \Big(\sum_{j=0}^{N} \mathrm{Re}(y_j - \hat{y}_j)^2 + \mathrm{Im}(y_j - \hat{y}_j)^2\Big)^{0.5} \quad (5)$$

where $N$ is the amount of datapoints, $y_j$ is the value of the $j$th datapoint and $\hat{y}_j$ the fit value for the given datapoint. The smaller the distance of the fit from the data, the better the fit. The measure provided in Eq. 5 is akin to a variance, whereas the measure provided in Eq. 4 is less disturbed by outliers. These distances were calculated for each condition using the fits generated with python 3.6 using the module symfit[47]. As fit models we used the single power-law $\hat{y} = A\left(\frac{f}{f_0}\right)^\alpha$, double-power law model, structural damping model and Maxwell model (for equations see Supplementary Fig. 7).

To test for a correlation between the cell radius $R$ and the cell mechanical properties $E'_{\mathrm{cort}}$ and $E''_{\mathrm{cort}}$ we performed a $t$-test on the data shown in Fig. 4c, d. The null hypothesis $H_0$ for the test is $H_0 : c = 0$, with $c$ being the correlation coefficient, i.e., $H_0$ is no correlation. A $p$-value $<0.05$ indicates a rejection of this null hypothesis, at higher $p$-values $H_0$ is not rejected. This means, if the $p$-value is above the significance level there is no evidence for the alternative hypothesis, i.e., correlation with $R$.

**Chemical perturbations.** Chemicals used to perturb HeLa cells were dissolved in dimethyl sulfoxide (DMSO), then diluted in cell culture medium. 30 min before the measurements, the solution was added to the medium in the cell culture chamber, containing the cells. The final concentrations were 500 nM for latrunculin-A (LatA, Sigma L5163) and 50 μM for CK666 (Sigma SML0006).

**Stimulated emission depletion (STED) nanoscopy to estimate cortex thickness.** HeLa cells in a Petri dish were fixed with 4% paraformaldehyde and stained using a SiR-actin kit (Spirochrome; SiR-actin 1:1000) mounted on an inverted Axio Observer Z1 (Zeiss), equipped with an a-Plan Apochromat 100x/1.46 NA oil objective (Zeiss) and a STEDYCON super resolution module (Abberior Instruments). Single cells were selected using low-resolution scans and super resolution $z$-slices were recorded at the cell center. Super resolution images were acquired with a power of 1.8 μW for the 640 nm stimulation laser and 0.16 W for the depletion laser, with a maximum lateral resolution of 50 nm. Images were acquired at line scan frequency of 42 Hz and pixel dwell time of 1 μs. Each line was scanned five times and fluorescence signals accumulated. Acquired images were subjected to Huygens professional deconvolution (v19.04, Scientific Volume Imaging) using the theoretical point spread function of the objective.

To assess cortex thickness, STED images were background subtracted and line profiles at random cortex positions were acquired using ImageJ (Wayne Rasband). For each cell, 10 line profiles (fluorescence intensity $I$ vs position $s$) were extracted at different locations. The line profiles were fitted with a Gaussian

$$I = I_0 + I_1 \cdot \frac{\exp\left[-\frac{4\ln(2)(s-s_0)^2}{w^2}\right]}{w\sqrt{\frac{\pi}{4\ln(2)}}} \quad (6)$$

where $I_0$ is the signal offset, $I_1$ the area of the fluorescence intensity peak, $s_0$ the center position of the peak, and $w$ the full width at half maximum. Using the average parameter $w$ of Eq. 6 as cortex thickness of a single cell, we extracted values of 10 individual cells and determined the mean and standard error (Supplementary Fig. 15).

To measure the cell radius $R$ for the subsequent correlation with the cortex thickness, a line was drawn through the center of the cell, crossing the cell cortex on two opposite sides. The distance of the fluorescence intensity peaks on either side was taken as a diameter. A second diameter was measured for the same cell using a line orthogonal to the first one. Averaging the two diameters and halving the result, gave the reported $R$.

**STED nanoscopy to estimate the shape of compressed cells.** For $xz$-slices of compressed HeLa cells the STED setup (see above) was equipped with a 63x water-immersion objective (LCI Plan-Neofluar 63x/1.3 Imm Corr DIC M27, Zeiss). The setup was placed in an environmental control chamber (The Cube, Life Imaging Services), to maintain ambient temperature at 37 °C. As previously described, microcantilevers (ACL-TL-10; AppNano) were sculpted into wedged micro-cantilevers by a focused ion beam to compensate for the AFM-based system

intrinsic 10° tilt of cantilever and to allow parallel plate compression of HeLa cells[13].

HeLa cells cultured as described above were suspended in STED-medium (DMEM (12800017, Thermo Fisher) supplemented with 20 mM HEPES (A3724, Applichem)), supplemented with 1 % FCS, pelleted and resuspended in STED-medium. HeLa cells were allowed to recover from trypsin treatment for 30 min, before being pipetted onto glass-bottom Petri dishes (WPI) containing STED-medium supplemented with 1 μM SiR-actin and 10 μM Verapamil (both spirochrome). Appropriate HeLa cells, that showed a minimal contact area to the glass-bottomed petri dish were selected for imaging. The laser intensity of the excitation laser was optimized for each cell to gain best signal-to-noise images. The depletion laser was set for a $xy$-resolution of 100 nm. For each selected HeLa cell two perpendicular and cell-centered $xz$-slices were acquired. Afterwards, the wedged cantilever was approached to the HeLa cells until a relative deflection setpoint of 0.05 V was recorded. Subsequently, the cantilever was lowered 1 μm and for compressed HeLa cells two perpendicular and cell-centered $xz$-slices were acquired with the same settings as uncompressed.

**Simulation of cell stiffening.** For the simulations shown in Fig. 1e, we used the equations of motion based on the spring-damper model (inset, Fig. 1e):

$$k_\mathrm{m}x_\mathrm{m} + c_\mathrm{m}\dot{x}_\mathrm{m} - k_\mathrm{c}(x_\mathrm{s} - x_\mathrm{m}) + m_\mathrm{m}\ddot{x}_\mathrm{m} = F_m \quad (7)$$

$$k_\mathrm{s}x_\mathrm{s} + c_\mathrm{s}\dot{x}_\mathrm{s} - k_\mathrm{c}(x_\mathrm{m} - x_\mathrm{s}) + m_\mathrm{s}\ddot{x}_\mathrm{s} = 0 \quad (8)$$

We solved the Eqs. 7 and 8 using MatLab R2016b. The variables are labeled according to Fig. 1c. Except for $k_\mathrm{c}$, which was chosen to be 3 mN m$^{-1}$, the other constants were from the experiment. To simulate the hardening of the cell, $k_\mathrm{c}$ was incremented up to 100-fold at the final time point.

**Comsol solid-shell–liquid-core cell model.** Due to the small dynamic compression by the confining microcantilevers, it is assumed that the cell can be modeled with the solid-shell–liquid-core model. For the modeling, the Comsol solid mechanics module (Comsol Multiphysics GmbH) was utilized in a 2D axial symmetric mode. The initial form of the cell is assumed to be spherical, its radius $R$ being calculated from the user-given cell volume $V_\mathrm{c}$ (Supplementary Note 2). The shell was parameterized with the cortex thickness $h_{\mathrm{cort}} = 200$ nm. Axial symmetry is utilized, together with constraining the vertical displacement of the lower boundary of the cell. The zero vertical deformation condition acts as a $zy$-horizontal symmetry plane. Therefore, the cell will be squashed symmetrically from top and bottom by the same amount. The fluid inside the cell is assumed to be incompressible, therefore the internal volume of the modeled cell is kept fixed for every compression distance $\Delta$. This is done via a global equation: $\oint(R + u(P)) \cdot nA_\mathrm{c} - V_\mathrm{c} = 0$. This equation, solves the internal pressure $P$ such that the internal volume, calculated from the deformed cell shape $u(P)$, matches $V_\mathrm{c}$ for every $\Delta$. From the initially spherical shape, the cell is pressed vertically with the help of a contact load. Instead of utilizing contact surfaces in Comsol, we modeled the load by specifying a contact force per unit area of the form $F_A \propto \theta(x - R + d\Delta/2)$, where $\theta$ is the Heaviside unit step function, and $d\Delta$ is a user-controlled parameter describing the compression of the cell. Finally, the force acting on the cell can be obtained by integrating the vertical reaction force in the contact area. The spatial derivative of the force, gives the spring constant as function of the compression distance of the cell, which is used to correlate spring constant with cell stiffness (Supplementary Note 2).

**Comsol cantilever–cell–cantilever system model.** The entire cantilever–cell–cantilever system was modeled using the Comsol solid mechanics module. The cantilever geometry was designed in SolidWorks 2018 (Dassault Systèmes SolidWorks Corp.) and imported into Comsol. The cell was modeled as a compressed solid sphere and is meshed with free tetrahedra (Fig. 2e, g). The top surfaces of the cantilevers were first meshed with free triangles and then swept into the thickness. Force was applied to the cantilevers as a uniform edge load on transverse line cuts at $z = 20$ μm (Fig. 2b). The cantilevers are modeled as silicon nitride (Si$_3$N$_4$, 250 GPa storage modulus) and having phenomenological internal damping that matches the quality factor of the microcantilever in fluid. The density of Si$_3$N$_4$ was adjusted to match the resonance frequency of the microcantilever in fluid. The cell was simulated to be homogeneous, with a frequency-dependent storage modulus described by either a double power-law model, a structural damping model, a Maxwell model, a polynomial function or an oscillating polynomial function (Supplementary Fig. 7). To extract the response functions for configurations $MM$, $MS$, $SS$ a full frequency domain simulation was performed. The vertical displacement was sampled at the free end of each cantilever ($z = 110$ μm) and plotted versus the actuation frequency (Fig. 2).

## Data availability
Data supporting the findings of this manuscript are available from the corresponding author upon reasonable request. A reporting summary for this Article is available as a Supplementary Information file. Source data are provided with this paper.

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

## Acknowledgements

We thank Ch. Gerber and J. Helenius for fruitful discussions as well as P. Argast for manufacturing wedges. We thank S. Schmitt for providing alignate beads. This work was supported by the NCCR Molecular Systems Engineering, Innosuisse (project 28033.1 PFNM-NM), and Swiss Nanoscience Institute Basel.

## Author contributions

G.F., D.M.-M., and D.J.M. discussed the experimental setup. G.F. set up the rheology experiment, designed and conducted the experiments, developed the theory, and analyzed the data. G.F., D.M.-M., N.S., and D.J.M. evaluated the experimental progress and data. G.F., N.S., and D.J.M. designed follow up and control experiments and revised the paper. C.I.R. and G.F. performed the simulations. N.S and G.F. conducted super-resolution microscopy. All authors discussed the experiments, wrote, read, and approved the manuscript.

## Competing interests

The authors D.M.-M, D.J.M., and G.F. declare the following competing interests: D.M.-M. and D.J.M. have filed two patents related to the technology of the microoscillator-based device for mass and mechanical measurements (PCT/EP2015/000350) and its applications (EP3108283A1). D.M.-M., G.F., and D.J.M. have filed a patent related to the environmental chamber (PCT/EP2016/001243). C.I.R. and N.S. declare no competing interests.
