## [Peer Review File · Nature Communications]

Reviewer #1 (Remarks to the Author):

The authors have addressed my concerns. I recommend publication.

Reviewer #2 (Remarks to the Author):

Overall, this work introduces and demonstrates a useful new tool in cell rheology. The manuscript should be publishable after addressing a few points, as outlined below.

As I noted before, the basic approach here should be viewed as a valuable and publishable extension or alternative to prior methods. The authors point to several “several (compelling) cases [of] new biophysics” that can be learned by their method. As outlined in the (prior and current) abstract, these new insights are the three listed in my prior review. I indicated that these will not come as a surprise to readers familiar with the prior literature in this field. The authors take issue with this, suggesting that these results must be shown experimentally and implying that such experimental evidence is lacking. While I accept that I may have been overly dismissive of one of these, I still find that the authors overstate the other two.

My comment about the lack of surprise was because at least the first two of these HAVE been shown experimentally. On point (1), there are now a number of papers concluding that active/regulated aspects of mechanics are limited to approximately 10s of Hz at most. See, e.g., Ref. 10 and Guo et al, Cell (2014). These constitute more direct methods to “dissect” unregulated (passive) from regulated (active or non-equilibrium) aspects of mechanics, since they can test non-equilibrium conditions.

In this regard, I find the authors implied claims of novelty to be somewhat overstated. For instance, in lines 41-43, they say that “a qualitative convergence of passive and active rheological approaches has been observed in some [cases],” where they cite Ref. 10. The experiments in Ref. 10 are just a quantitative as the present ones, and the convergence there and by Guo et al (Cell, 2014) is demonstrated quantitatively. The authors go on to say that “a universality of this convergence has not been reported.” While the prior results may not have been claimed as “universal”, I am unaware of any experimental results demonstrating a lack of such convergence at high frequencies above 10-100 Hz. It is fair to say that it is generally accepted following a number of experiments that (biological) activity/regulation is only apparent at lower frequencies. Thus, I find the implication that the authors are filling a significant void here to be overstated.

Elsewhere, I find the implication from the introductory discussion that prior results have been limited to ~100 Hz (line 39), while they achieve 40kHz (line 57) to misstate the current state of the art. In fact, while their raw data seem to extend almost to 40kHz (e.g., Fig. 1d), the actual rheological data on cells (e.g., for E in panels b-d of Fig. 3) are really limited to about 25kHz, since the data for the last few kHz are too noisy to interpret. By comparison, Ref. 10 extended to ~8kHz. Thus, I would say that the present method extends prior approaches by about one-half decade. As I noted, this work introduces a valuable tool for high frequency rheology. But, the authors should be more

accurate and fair in their characterization of the prior state of affairs.

On point (2), there are even more papers in the literature showing power-law rheology (e.g., Refs. 8 & 24). So, as I said, this aspect will not be a surprise to readers familiar with prior work in the field. In their response, the authors point to the different geometry here (rounded vs spread cells). This is a fair point, as the prior work on such power-laws has (exclusively, to my knowledge) been on adherent cells. So, while the present work shows general consistency with prior work on adherent cells, the results are new for rounded cells.

With referee 3, I also do not see the final point (3) about the Laplace law as a major insight. At best, this may correct a few people in the field. From a (bio)physical perspective, I would still say there is nothing surprising in the inconsistency with Laplace.

Finally, I am a bit confused by the authors' reply to the other referee's comment on inertia (3rd portion of reply). The authors seem to suggest that the only possible source of inertial effects is poroelasticity. While I agree with the overall conclusion that inertial is likely not important on timescales in excess of a few microseconds, even simple Newtonian liquids that are not poroelastic exhibit inertial effects. A more relevant timescale to my mind is $d^2 \rho / \eta$, corresponding to the diffusive propagation of vorticity with diffusion constant η / ρ , where η is the viscosity and ρ the density of the fluid. For viscoelastic fluids, the propagation is superdiffusive and will depend on the frequency-dependent modulus or viscosity. Anyway, again, I agree that inertia is likely not relevant on the timescales here, particularly in view of the larger moduli/viscosity of cells.

Reviewer #3 (Remarks to the Author):

Although I don't necessarily agree with all the responses to my concerns, I think that the manuscript is now acceptable for publication.

Point-by-Point Response to the Reviewer's Comments

Point-by-Point Response to Reviewer #1

Reviewer #1: The authors have addressed my concerns. I recommend publication.

Authors: We thank the reviewer for his/her critical and constructive comments, which have guided us to improve our manuscript.

Point-by-Point Response to Reviewer #2

Reviewer #2: Overall, this work introduces and demonstrates a useful new tool in cell rheology. The manuscript should be publishable after addressing a few points, as outlined below.

Authors: We thank the reviewer for his/her critical and constructive comments, which have guided us to improve our manuscript. In our revised version we have addressed the issues brought up, in particular we toned down the claims of novelty.

Reviewer #2: As I noted before, the basic approach here should be viewed as a valuable and publishable extension or alternative to prior methods. The authors point to several "several (compelling) cases [of] new biophysics" that can be learned by their method. As outlined in the (prior and current) abstract, these new insights are the three listed in my prior review. I indicated that these will not come as a surprise to readers familiar with the prior literature in this field. The authors take issue with this, suggesting that these results must be shown experimentally and implying that such experimental evidence is lacking. While I accept that I may have been overly dismissive of one of these, I still find that the authors overstate the other two.

Authors: Thank you. In our revised version we have particular we toned down the claims of novelty.

Reviewer #2: My comment about the lack of surprise was because at least the first two of these HAVE been shown experimentally. On point (1), there are now a number of papers concluding that active/regulated aspects of mechanics are limited to approximately 10s of Hz at most. See, e.g., Ref. 10 and Guo et al, Cell (2014). These constitute more direct methods to "dissect" unregulated (passive) from regulated (active or non-equilibrium) aspects of mechanics, since they can test non-equilibrium conditions.

In this regard, I find the authors implied claims of novelty to be somewhat overstated. For instance, in lines 41-43, they say that "a qualitative convergence of passive and active rheological approaches has been observed in some [cases]," where they cite Ref. 10. The experiments in Ref. 10 are just a quantitative as the present ones, and the convergence there and by Guo et al (Cell, 2014) is demonstrated quantitatively. The authors go on to say that "a

universality of this convergence has not been reported.” While the prior results may not have been claimed as “universal”, I am unaware of any experimental results demonstrating a lack of such convergence at high frequencies above 10-100 Hz. It is fair to say that it is generally accepted following a number of experiments that (biological) activity/regulation is only apparent at lower frequencies. Thus, I find the implication that the authors are filling a significant void here to be overstated.

Authors: Thank you again. We have revised our manuscript and deleted our claims about the inability of passive approaches to characterize non-equilibrium systems (see revised Manuscript, first paragraph).

Reviewer #2: Elsewhere, I find the implication from the introductory discussion that prior results have been limited to ~100 Hz (line 39), while they achieve 40kHz (line 57) to misstate the current state of the art. In fact, while their raw data seem to extend almost to 40kHz (e.g., Fig. 1d), the actual rheological data on cells (e.g., for E in panels b-d of Fig. 3) are really limited to about 25kHz, since the data for the last few kHz are too noisy to interpret. By comparison, Ref. 10 extended to ~8kHz. Thus, I would say that the present method extends prior approaches by about one-half decade. As I noted, this work introduces a valuable tool for high frequency rheology. But, the authors should be more accurate and fair in their characterization of the prior state of affairs.

Authors: Thank you. We apologize for the misunderstanding our phrasing has led to. When stating that “So far, low and intermediate frequency ($10^{-3} - 10^2$ Hz) recordings predominate because they are easily approachable. Higher frequencies are accessible by passive rheology approaches...” we did not wish to imply that there are no higher frequency approaches in general, but that the *active* rheology (where force is applied to the sample) is typically rather restricted. In fact, Ref. 10 itself extends to ~ 8 kHz, as the reviewer states, but only for the *passive* approach (open symbols in Fig. 2 of the respective paper), where the *active* rheological measurement (filled symbols in Fig. 2 of respective paper), extends to ~ 100 Hz. We find it to be common, that active approaches, do not reach the frequencies of passive approaches. We have rephrased the sentences in hope to avoid any misunderstanding (see revised Manuscript, first paragraph).

Reviewer #2: On point (2), there are even more papers in the literature showing power-law rheology (e.g., Refs. 8 & 24). So, as I said, this aspect will not be a surprise to readers familiar with prior work in the field. In their response, the authors point to the different geometry here (rounded vs spread cells). This is a fair point, as the prior work on such power-laws has (exclusively, to my knowledge) been on adherent cells. So, while the present work shows general consistency with prior work on adherent cells, the results are new for rounded cells.

Authors: We thank we author for considering our argument. We have rephrased the main manuscript such it becomes evident, that novelty lies in the power-law behavior within the context of rounded cells and not in the power-law behavior by itself. (see revised title of manuscript, abstract and second paragraph).

Reviewer #2: With referee 3, I also do not see the final point (3) about the Laplace law as a major insight. At best, this may correct a few people in the field. From a (bio)physical perspective, I would still say there is nothing surprising in the inconsistency with Laplace.

Authors: We understand the position of the reviewer and have toned down our claim on the importance of this finding (see revised Manuscript, abstract and first paragraph).

Reviewer #2: Finally, I am a bit confused by the authors' reply to the other referee's comment on inertia (3rd portion of reply). The authors seem to suggest that the only possible source of inertial effects is poroelasticity. While I agree with the overall conclusion that inertia is likely not important on timescales in excess of a few microseconds, even simple Newtonian liquids that are not poroelastic exhibit inertial effects. A more relevant timescale to my mind is $d^2 \rho/\eta$, corresponding to the diffusive propagation of vorticity with diffusion constant η/ρ , where η is the viscosity and ρ the density of the fluid. For viscoelastic fluids, the propagation is superdiffusive and will depend on the frequency-dependent modulus or viscosity. Anyway, again, I agree that inertia is likely not relevant on the timescales here, particularly in view of the larger moduli/viscosity of cells.

Authors: We thank the reviewer for this insightful remark!

Point-by-Point Response to Reviewer #3

Reviewer #2: Although I don't necessarily agree with the all the responses to my concerns, I think that the manuscript is now acceptable for publication.

Authors: We thank the reviewer for his/her comments which have guided us to revise our manuscript.